# MERLIN: Multiple Enhanced Representations with LLM Generated INdices

**Anirudh Ravichandran***
aniravic@amazon.com
Amazon
Seattle, WA, USA

**Yidong Zou***
yid@amazon.com
Amazon
Seattle, WA, USA

**Jayapragash Baskar**
jbaskar@amazon.com
Amazon
Seattle, WA, USA

**Anurag Beniwal**
beanurag@amazon.com
Amazon
California, CA, USA

## ABSTRACT

Large Language Models(LLMs) can be leveraged to improve performance in various stages of the search pipeline - index enhancement , query rewriting, and ranking or re-ranking. The latter two methods involve using large language model calls during inference, adding latency in fetching the final ranked list of documents. Index enhancement, on the other hand can be done in the indexing phase in near real time, and can result in improved retrieval performance while adding no or minimal additional latency during query-time inference. Enhancing indexes with information generated by LLMs is a promising mechanism to improve first stage retrieval results in dense retrieval using bi-encoders, on par or exceeding the other two approaches. In this work, we show that by using multiple indexes to represent documents in different ways, where the representations are generated by an LLM, and querying these indexes in parallel, we can improve retrieval performance with almost no increase in runtime latency. Our results are consistent across a number of pre-trained bi-encoder models. We detail the implementation of such a system in an industrial setting with AWS services in the customer service domain to help retrieve the correct self-help content for an amazon customer query.

## CCS CONCEPTS

• **Information Systems** → **Information Retrieval**.

## KEYWORDS

Retrieval models, Large Language models

## 1 INTRODUCTION

Amazon customer service serves customers that contact Amazon through various self service and automation tools. One of the fundamental applications in Amazon customer service is the document retrieval service, which retrieves relevant content in response to a customer query. Typically, a search pipeline involves 4 phases -

---

*Both authors contributed equally to the paper.

1. The indexing phase, where the documents from the corpus are indexed. 2. The query understanding phase, where the query can be expanded, transformed etc. 3. Scoring or retrieval phase, where the document and query representations interact with each other to generate scores in order to retrieve an initial list of candidates. 4. A re-ranking phase, where the results from phase 3 are further refined. Recent work has shown that LLMs can be used in the query understanding phase [3, 6, 11] to generate various runtime transformations for the original query that help retrievers improve their performance. For example, HyDe [3] transforms the query to the document space by rewriting the query into a hypothetical document. Similarly, Query2Doc [6, 11] generates runtime expansions for the query with relevant terms in order to enrich the query. LLMs have also been shown to be effective as the "reranker" to help successfully re-rank results from the first stage retrieval in response to a query [10]. These methods, while effective, have the downside of involving an LLM call during query-time inference which is both computationally expensive and incurs high latency.

In an industrial grade applications such as customer service AI systems, where multiple models may be present in cascade, methods that leverage LLMs in a latency-friendly manner are desirable. With RAG applications and vector search solutions exploding in popularity, we believe that the use of LLMs to enhance and generate expansion terms for documents during indexing time has been under-explored. This work discusses the use of LLMs to enhance document indexes offline to improve retrieval performance. It also provides the details of the setup and deployment of such a near-real time system using AWS services. We show that by prompting LLMs to generate various representations for documents, and maintaining multiple indices for documents and querying them in parallel, we are able to improve first-stage retrieval performance while incurring minimal additional latency. We compare our approach to those that use LLMs in the query understanding phase, and in the re-ranking phase. In both cases, we observe comparable or superior performance of multi-index enhancement approaches but with the added advantage of no or minimal additional latency. We present both offline and online results for such a system that indexes publicly available help content available on amazon.com.

Our main contributions are as follows:

1. We demonstrate that by leveraging LLMs to enhance and maintain multiples indexes, we can get improvements in first stage retrieval performance. Our results are consistent across multiple bi-encoder models and various prompting strategies.

2. We detail the implementation of such a system using AWS services and thus provide a template for a multi index system that improves retrieval in an industrial setting.

## 2 METHODOLOGY

### 2.1 Embedding Model and Dataset Details

For the purpose of demonstrating the effectiveness of our approach, we fine-tune bi-encoder models[8] for the customer service domain and use that across all our approaches as the first stage retrieval. All experiments are performed on publicly available self-help documents on the customer service tab on Amazon.com (example: Track Your Package). Because these articles are far fewer than in a web-scale setting, we use only a single stage of retrieval and some lightweight re-ranking as detailed in section 2.3.

In order to fine-tune the model, we generate around 30K data points in a self-supervised fashion by using the title of the document and the body content of the document as a positive pair. To this, we also include 10k queries mimicking the real customer input (and the corresponding ground truth document) that were manually labeled. In order to gather ground truth relevance information, we presented the annotators with the query and the top-6 options as picked from an LLM from an initial candidate pool of 15 candidates that was a generated by a previously optimized BM25-style system. It is worth noting that some of our annotators have previously worked as Customer Service Representatives, and so are domain experts in identifying the right document for a given customer query.

We train the biencoder embedding model using the Multiple-Negative ranking loss [5]. Our test set consists of 5K "difficult" queries (and a single relevant document as a ground truth answer) as rated by human annotators, that are representative of real customer queries to Amazon. Throughout the remainder of the paper, we present results with the MP-net model [9] that was pretrained on 1B sentences and is optimized for semantic search/relevance (Huggig Face model ID: sentence-transformers/all-mpnet-base-v2), though our results are consistent across a variety of bi-encoder models.

### 2.2 Multi-Index enhancement

Inspired by previous work [2], we hypothesize that for dense retrieval, the presence of varied, but highly relevant terms that represent the main themes in the document can generate a more representative embedding than feeding in the entire document since the document may also include individual sentences that are unhelpful or not representative of the theme of the overall document which may perturb the embeddings generated by Bi-encoder models. Accordingly, we prompt an LLM model to produce separate 5-6 sentence and 2-3 sentence summaries for every document in

the index. Also, inspired by past work [7], we recognize the importance of having additional expansion terms in the document to help BM25/biencoder models match the right document. We also prompt the LLM to generate 5 queries in a doc2query style [4] and 5 high-level labels or "tags" for the document.

After experimenting with various index-forms, we empirically zeroed-in on the following 5-index setup:

1) *Content Index*: This is the original HTML content of the document.

2) *LLM Summary Index*: In this index, the LLM generates a concise summary of the document in 6 sentences and that summary gets indexed.

3) *LLM Short Summary Index*: In this index, the LLM generates a highly concise summary of the document in 3 sentences and that summary gets indexed.

4) *LLM Questions and Tags Index*: In this index, the LLM generates 4 key questions that can be answered by the document in the doc2query style, and also 4 keywords or phrases that can describe the document and these get indexed.

5) *Metadata Index*: In this index, the document category information get indexed along with the title of the article.

Each entry in every index gets prefixed with the title of the document. We've found the title of the document to be the single most powerful signal in generating a representative embedding for the document. This also follows from our self-supervised training approach highlighted in the previous section.

### 2.3 Combining Results from Indexes

Reciprocal rank fusion [1] gives us a way to combine ranks from different indexes based on a document's rank in individual indices. We use a variant of reciprocal rank fusion that also takes into account the presence of a document in the top-5 positions for that index - we use the following empirical formulate to score the documents. The final score for a document $j$ can be calculated as:

$$score_j = (\sum_{i \in I} \frac{sim_{ij}}{rank_{ij}}) * frac_j$$

where $frac_j$ is the fraction of indexes in which the document $j$ appears in the top-5 results, $sim_{ij}$ is the cosine similarity score between the embedding for the query and the document $j's$ representation in index $i$, $I$ represents the set of all indices and $rank_{ij}$ represents the rank of document $j$ in index $i$.

## 3 RESULTS

Recall performance is listed in Table 1 and the relative lifts in recall@k for various values of k and the different methods are presented in fig 2. As with re-ranking using LLMs, when relevance identification is offloaded to the LLM, as expected, an LLM is highly capable of identifying the top-1 and the top-2 results from the list of top-5 results, however we observe diminishing gains for k>2. This suggests that LLMs, while good at identifying the best of top-2 best documents from a list, don't particularly do well at exhaustively ranking a list of results.

# Indexing Architecture

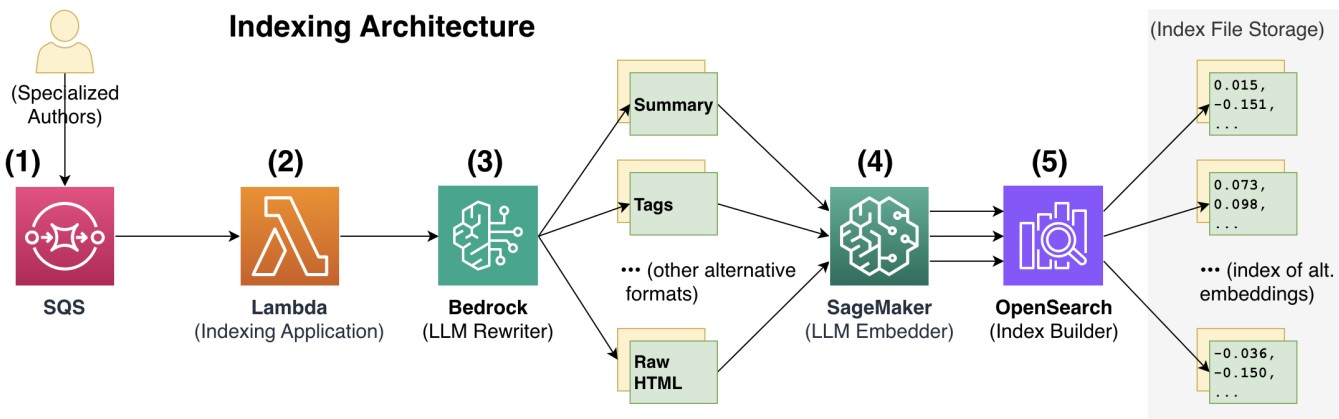

**Figure 1: Software architecture of the indexing component**
**(1) AWS SQS queue collects newly modified documents (2) AWS Lambda processs the queued documents and orchestrates indexing steps 3-5 (3) LLM in AWS Bedrock rewrites each input document in multiple forms (4) LLM model in AWS SageMaker produces an embedding for each document rewrite (5) AWS OpenSearch builds an index for each style of rewriting**

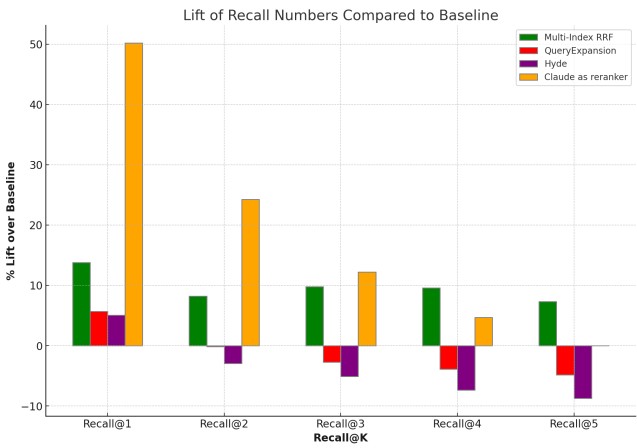

**Figure 2: Recall@K performance comparison**

| Relative lift in Recall@k (%) over the baseline | | | | | |
|---|---|---|---|---|---|
| Method | k=1 | k=2 | k=3 | k=4 | k=5 |
| HyDe | 5.07 | -2.97 | -5.13 | -7.37 | -8.74 |
| Query2Doc-CoT | 5.67 | -0.17 | -2.74 | -3.87 | -4.85 |
| Reranking-CoT | **50.14** | **24.27** | **12.22** | 4.66 | 0.00 |
| Multi-Index RRF (ours) | 13.78 | 8.21 | 9.78 | **9.55** | **7.31** |

**Table 1: Relative Recall@K improvements of various methods**

We find that our best performing single-index for dense retrieval is indeed the short summary index that is produced by the LLM, introducing highly relevant terms and brief sentences that are highly representative of the high level semantic concepts in the document. As an example, consider the query *'i am currently in school and would like to try the amazon membership.'* Dense retrieval over the default content index (baseline) ranks the 'Prime student' document

and 'Join Prime student' document in 1st and 3rd place respectively. It ranks 'Sign up for Amazon Prime' at the 2nd place. When matched with the short and long summary, the 'Join Prime student' correctly ranks in 1st place with a much higher score. This document also ranks 1st in the *Questions and tags index* and consequently is ranked 1st in the multi index setting. The text encoded as part of the Join Prime Student document across all indices is shown below. The embedding based on the summary of the document, and its tags and questions is much more representative of the concept "joining prime student", while in contrast, the embedding obtained from the raw document content is closer to the concept "verify student status" because of the nature of the exact text in the document. The text in the document actually contains quite a few sentences dedicated to verifying student status even though the high-level goal is to list steps to join Amazon Prime Student. Here is the example of the above self-help content represented across multiple indexes:

*Content Index:* Content at Join Prime Student

*LLM Summary:* Students can sign up for a free 6-month trial of Amazon Prime Student membership. To join, students go to the Prime Student website, fill out a form, and verify their email. The trial includes free 2-day shipping and other benefits like streaming media, music, photo storage, and discounts. Students can use a non-.edu email but may need to resend a verification email. Current Prime members get a refund for remaining time and Prime Student benefits cannot be shared.

*LLM Short Summary:* The article explains how to sign up for a free trial of Amazon Prime Student, which is Amazon's prime program tailored for students enrolled in college. It includes free 2-day shipping and other benefits like free streaming and gaming perks. To sign up, students need to verify their status by submitting proof of enrollment, a .edu email address, or documents

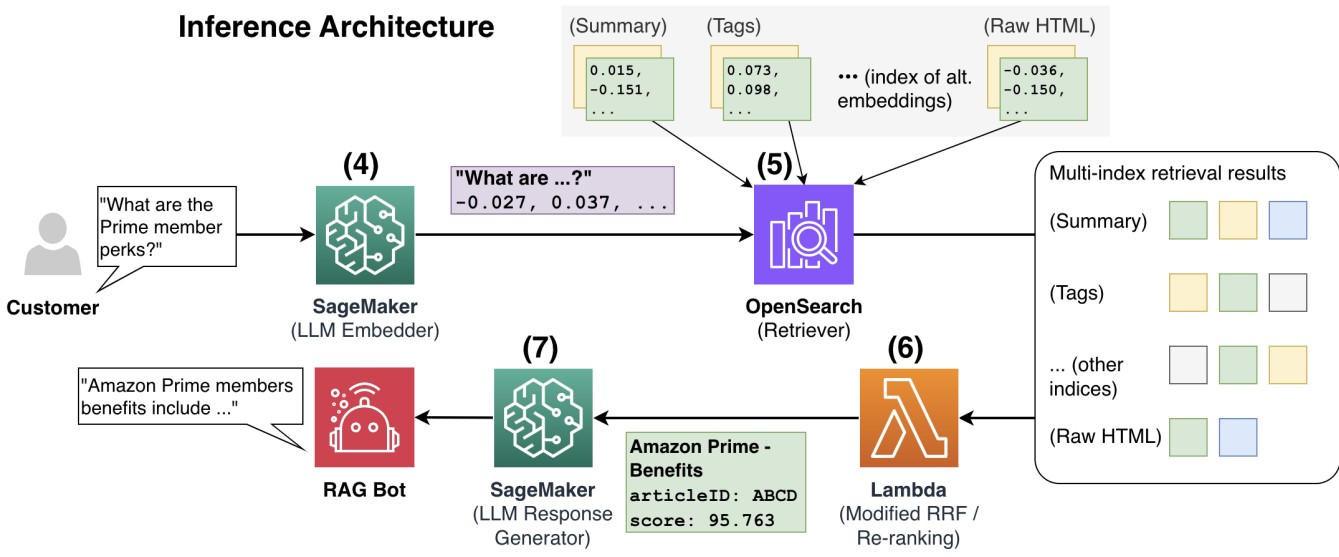

**Figure 3: Software architecture of the inference component (4) LLM model in AWS SageMaker produces an embedding for the input customer query (5) AWS OpenSearch retrieves candidate documents from multiple document indices (6) AWS Lambda re-ranks all the top candidates using algorithms such as modified RRF (7) LLM model in AWS SageMaker paraphrases the best matching document into a coherent answer**

showing their age.

*LLM questions and tags:* Queries: 'How do I sign up for Amazon Prime Student?', "I'm a student, can I join Amazon Prime?", 'Is there a student discount for Amazon Prime?', 'I am in college and I would like to trial out the Amazon Student Prime membership program', 'tags': 'Prime Student Membership', 'Student Prime', 'Amazon Prime for Students', 'Sign up for Amazon prime student'

**Simulated Online Results** - In an experiment simulating a week-long traffic on amazon.com, we observed a 0.25% decrease in average agent Contact Per Customer (CPC) in response to improved ranking results for customer queries. Contact Per Customer (CPC) is an important business metric that stands as one of the proxies for the effectiveness of our automated solutions, ranked self help content being one of them.

### 3.1 Comparison to other approaches

We compare our approach of index-enhancement to:
1. Using an LLM to augment the query at run-time by introducing expansion terms, *Query2Doc* [6, 11].
2. Using an LLM to transform the query to document space , *HyDE* [3].
3. Using an LLM to re-rank an initial set of results, *Reranker* [10].

For Query2Doc and Re-ranking, we use 5-shot CoT prompting with an LLM to generate query expansion terms and re-ranked results. Multi-Index RRF (using 5-shot prompting) outperforms all the compared approaches for Recall@5 performance while being highly amenable to a production setting. Crucially, by re-using the query embedding across the indices and parallelizing AWS Open

Search calls to multiple indices, we achieve these gains at no extra latency (the latency overhead introduced by RRF/any heuristic re-ranking or filtering is negligible in this case). While re-ranking using an LLM shows superior recall@k performance for K<3, its performance is ultimately bounded by the quality of first stage retrieval. As such, each of the presented methods can be used in conjunction with the multi-index approach. In an industrial settings, there are strict constraints on latency for ranking/retrieval systems, rendering methods like HyDE or re-ranking with LLMs infeasible (refer the qualitative plot in fig 4). With 50B+ LLMs, it can consistently take over 0.8 sec to produce query transformations/ re-ranking with CoT which may not satisfy the latency constraints. The multi-index approach makes documents available to query in near-real time and once indexed, adds nearly no additional latency cost during query-time inference.

### 4 SYSTEM DETAILS

System details during indexing time and query time are shown in figure 1 and figure 3 respectively. To keep the documents up-to-date with Amazon's latest policies and offerings, a team of specialized authors in Amazon Customer Service compose new or update existing documents. With the help of AWS SQS and AWS Lambda, we fetch these document changes (including creation, update, and deletion) in a near-real-time fashion with a configurable delay in minutes (Step 1). Upon receiving a change, we enrich the index by sending the document to an LLM hosted in AWS Bedrock to produce the various forms of rewrite. With parallel invocation, it takes about 3 seconds for the LLM to rewrite a single document (Step 3). We maintain a separate index for different forms of rewrite. The updated document, in multiple rewritten forms, are then converted into embeddings by the same model hosted in AWS SageMaker

Inference (Step 4). Note that this is not the exact system we use in production, and the diagrams here are for illustration purposes only to build out a system that is functionally capable of the components described.

The embedding model, which is shared across indices, creates one vector per document, per rewrite form, and the vectors are then collected into their respective indices with AWS Open Search (Step 5). Overall, the indexing micro-services make the new documents searchable in about 2 minutes. During inference, we compute the embedding of the customer query only once (Step 4), broadcast the query embedding across multiple Open Search indices, and retrieve the results in parallel (Step 5). Each of the parallel call returns its own ranking of candidate documents, and we truncate the ranking with a preset relevance score threshold to reduce noise from irrelevant documents during re-ranking.

## ACKNOWLEDGMENTS

We acknowledge the efforts from our partner engineering teams in prioritizing and enabling AWS Bedrock integration with our services to enable experimentation and thank them for their contribution.

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

# A  APPENDIX

## A.1  Latency-Performance Tradeoff Schematic

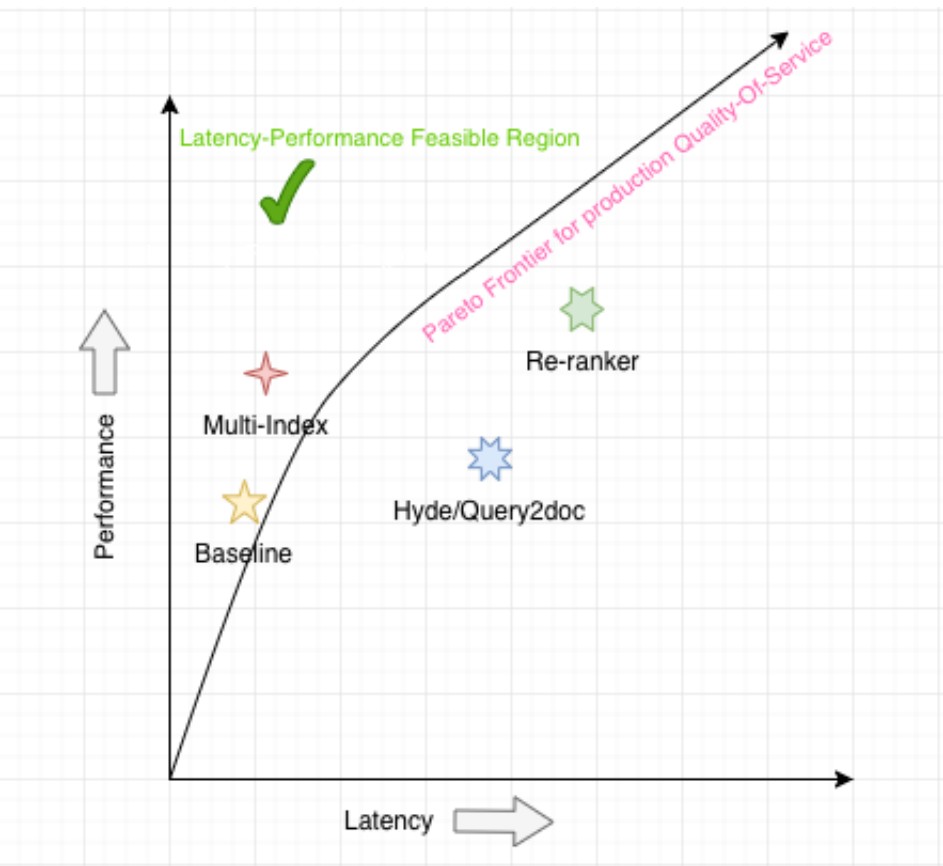

**Figure 4: Latency-Performance tradeoff schematic in the search pipeline**

Received 20 February 2007; revised 12 March 2009; accepted 5 June 2009