# OpenReview forum: "MERLIN: Multiple Enhanced Representations with LLM Generated INdices"
_ACM.org/SIGIR/2024/Workshop/Gen-IR — Gen-IR_SIGIR24_

### Official Review · Reviewer_VMRr · 2024-05-24

**Rating:** 1
**Confidence:** 4

**Review:**

This is a short paper that presents an industry application of applying generative models to not the retriever itself but to the construction of the index. i.e. in addition to the original index of documents, multiple parallel synthetic indices are constructed from various ways of rewriting the original document. During inference these indices are ensembled. This technique is able to keep inference cost low, as it is identical to normal dense retrieval (MIPS is cheap), while still introducing the benefits of generative models to the retrieval process.

Originality: This is a compelling (and to my knowledge novel) idea that's worth further exploration. In particular it is able to push the cost of query rewriting to the indexing stage, thereby saving inference cost.

Significance: The idea maybe significant but the exploration is limited in this paper, due to the industry focus (which is OK.) In particular, the paper would benefit from exploration into how this technique applies to a wider variety of retrieval tasks, or deeper study into the design choices in such a paradigm.

Quality & Clarity: The paper is generally decently written and clear but with a few typos and places where it could be clearer. Some things to point out:
- Miscapitalization in the title.
- It mentions Claude, but without clearly stating that it was the LLM used in the experiments.

Strengths:
- The idea is compelling and may be an effective and simple way to improve existing dense retrieval systems using generative models.
- The paper presents a system deployed to production, with end-user benefits.
- Especially beyond k@3, the method beats all other baselines, including generative reranking and query expansion via an LLM.

Weaknesses:
- The findings in the paper would be strongly supported by the inclusion of more ablations. e.g. how does each index contribute to the the overall performance? One index is a metadata index, how much of the result is from this metadata index vs. the synthetic indexes? Some discussion is included in the text but concrete numbers would be very valuable here.
- More discussion on limitations would be valuable. While this technique saves inference cost (no query rewriting or reranking), The task of generating summary or synthetic documents is more challenging and requires more expensive LLMs to run. Authors state a single LLM rewrite takes 3 seconds via Claude. Query rewriting and reranking could be done with much cheaper models. This pushes the cost of the technique to index building time, and may not scale to many documents without big cost.
- The online results could provide more detail. To what degree was the online experiment controlled for factors such as variance in time / traffic?
- This is not necessarily applicable due to the industry nature of this paper, but it would be great to see whether this technique applies to other retrieval tasks, and whether the LLM generations may introduce any biases.

---

### Official Review · Reviewer_CVEN · 2024-05-25
**Review: MERLIN: Multiple Enhanced Representations with LLM-generated Indexes**

**Rating:** 2
**Confidence:** 4

**Review:**

## Strengths

1. **Innovative Approach**: The use of LLMs to generate multiple enhanced representations for document indexes is a novel idea that addresses the limitations of existing methods involving LLMs in query understanding and re-ranking phases.

2. **Practical Implementation**: The detailed description of the implementation using AWS services provides a practical guide for deploying such a system in an industrial setting. This aspect enhances the paper's relevance and applicability.

3. **Latency-Friendly**: The method's ability to improve retrieval performance with minimal additional latency during query-time inference is a significant advantage, especially for applications requiring real-time responses.

## Weaknesses

1. **Limited Evaluation Scope**: While the results are promising, the evaluation is primarily focused on the Amazon customer service domain. Testing the approach on a broader range of datasets and retrieval tasks would strengthen the generalizability of the findings.

---

### Decision · Program_Chairs · 2024-05-30

**Decision:**

Accept

**Comment:**

## Decision: Accept

The authors propose an industry paper with insightful learnings for indexing a corpus with AWS services. The paper proposes harnessing some benefits of generative retrieval models at the latency cost of dense retrieval.
Additionally, the idea of using LLMs in this manner to index, rerank, and perform query understanding is considered novel by reviewers. The paper addresses several production-related details, such as latency, cloud services, and prompts."

The chairs are looking forward to further research on this topic and extensions to other cloud (agnostic) architectures, and sensitivity studies to different indexing strategies and retrieval tasks.

**For the camera-ready version, we ask the authors to consider the following changes**

- Explicitly mention Claude as the LLM used.
- If possible, a remark explicitely stating the cost tradeoff for this method versus a traditional dense retriever.
- If possible, elaborate on the online experiment setup.
- Depending on their preference, authors may consider the following capitalization for the title, or maintain the original one: 'Multiple Enhanced Representations with Llm Generated INdexes'.